# Normalizing HIF-1α Signaling Improves Cellular Glucose Metabolism and Blocks the Pathological Pathways of Hyperglycemic Damage

**DOI:** 10.3390/biomedicines9091139

**Published:** 2021-09-02

**Authors:** Carla Iacobini, Martina Vitale, Giuseppe Pugliese, Stefano Menini

**Affiliations:** Department of Clinical and Molecular Medicine, “La Sapienza” University, 00189 Rome, Italy; carla.iacobini@uniroma1.it (C.I.); martina.vitale@uniroma1.it (M.V.)

**Keywords:** carnosine, cellular energetics, diabetes, glycolysis, hyperglycemia, inflammation, methylglyoxal, prolyl 4-hydroxylase 2, trans-resveratrol, Warburg effect

## Abstract

Intracellular metabolism of excess glucose induces mitochondrial dysfunction and diversion of glycolytic intermediates into branch pathways, leading to cell injury and inflammation. Hyperglycemia-driven overproduction of mitochondrial superoxide was thought to be the initiator of these biochemical changes, but accumulating evidence indicates that mitochondrial superoxide generation is dispensable for diabetic complications development. Here we tested the hypothesis that hypoxia inducible factor (HIF)-1α and related bioenergetic changes (Warburg effect) play an initiating role in glucotoxicity. By using human endothelial cells and macrophages, we demonstrate that high glucose (HG) induces HIF-1α activity and a switch from oxidative metabolism to glycolysis and its principal branches. *HIF1-α* silencing, the carbonyl-trapping and anti-glycating agent ʟ-carnosine, and the glyoxalase-1 inducer trans-resveratrol reversed HG-induced bioenergetics/biochemical changes and endothelial-monocyte cell inflammation, pointing to methylglyoxal (MGO) as the non-hypoxic stimulus for HIF1-α induction. Consistently, MGO mimicked the effects of HG on HIF-1α induction and was able to induce a switch from oxidative metabolism to glycolysis. Mechanistically, methylglyoxal causes HIF1-α stabilization by inhibiting prolyl 4-hydroxylase domain 2 enzyme activity through post-translational glycation. These findings introduce a paradigm shift in the pathogenesis and prevention of diabetic complications by identifying HIF-1α as essential mediator of glucotoxicity, targetable with carbonyl-trapping agents and glyoxalase-1 inducers.

## 1. Introduction

The impact of hyperglycemia on the human vasculature is responsible for most of the morbidity and mortality in both type 1 and type 2 diabetes [1,2]. The injurious effect of hyperglycemia was attributed to biochemical consequences of intracellular metabolism of excess glucose. According to Brownlee’s unifying hypothesis, biochemical abnormalities are triggered by mitochondrial superoxide overproduction resulting from hyperglycemia-induced increase in electron donors from the tricarboxylic acid cycle. In turn, superoxide inhibits the glycolytic flux toward oxidative phosphorylation favoring the diversion of intermediates into glycolysis branch pathways, including polyol, hexosamine and advanced glycation endproduct (AGE) pathways [3]. However, at variance with previous reports showing increased superoxide levels [4], recent evidence suggests that superoxide production and mitochondrial function are decreased in the diabetic kidney [5,6]. Accordingly, the role of mitochondrial superoxide as the initial trigger of these ominous changes has been put into question [7,8]. Therefore, the unifying hypothesis of diabetic complications may need an alternative mechanism for instigating two main metabolic changes, i.e., (1) glucose-induced inhibition of respiration in the absence of superoxide overproduction; and (2) shifting of glucose flux from oxidative phosphorylation to the glycolytic pathway and its principal branches, ultimately leading to cell injury and inflammation.

Hypoxia-inducible factor-1α (HIF-1α) switches glucose metabolism from oxidative phosphorylation to glycolysis under hypoxic conditions (“Pasteur effect”) [9,10,11,12] by simultaneously increasing the expression of glycolytic enzymes and restraining mitochondrial function and oxygen consumption [13,14,15]. Under normoxia conditions, HIF-1α is rapidly hydroxylated by the oxoglutarate dependent prolyl 4-hydroxylase domain (PHD) 2 enzyme, which leads to HIF-1α proteosomal degradation [13]. The bulk of research on HIF-1α concerns crucial aspects of cancer biology [14]; hypoxia, which is a hallmark feature of the tumor microenvironment, promotes protein stability of HIF-1α subunit by inhibiting the activity of PHDs, as they require oxygen to hydroxylate HIF-1α [15]. However, even under aerobic conditions, most cancer cells tend to favor metabolism via aerobic glycolysis rather than through oxidative phosphorylation, a phenomenon termed “Warburg effect” [12,13,14,15]. In addition, metabolic changes resembling the Warburg effect play a role in inflammation, as increased HIF-1α signaling, reduction of glucose oxidation and switching to aerobic glycolysis are required for interleukin 1 β (*IL-1β*) mRNA induction in stimulated macrophages [16,17,18] and for myeloid cell-mediated inflammation [17]. Consistently, HIF-1α-deficient macrophages show a defective response to lipopolysaccharide (LPS) (i.e., loss of inflammatory capacity) [17]. Interestingly, hyperglycemia has also been reported to induce HIF-1α upregulation in renal and vascular cells [19,20]. Indeed, HIF-1α has been involved in vascular complications mainly because of the role of hypoxia in vascular injury and repair [21].

This study aimed at investigating whether high glucose (HG) conditions induce HIF-1α activity and related cellular energetic changes (i.e., Warburg effect), which may serve as initial mediators of glucose toxicity by activating the alternative pathways of glucose metabolism involved in hyperglycemia-induced vascular damage.

## 2. Material and Methods

This study was designed to test the hypothesis that HIF-1α plays an initiating role in HG-induced endothelial cell and macrophage activation by initiating a metabolic shift towards glycolysis and shunting of glucose into glycolysis side-branches.

### 2.1. Cell Culture

Human umbilical vascular endothelial cells (HUVEC), order number C-12203/C-12253, lot number 439Z032, were purchased from PromoCell (Heidelberg, Germany), human coronary artery endothelial cells (HCAEC), EGM-2MV, batch number 0000396592, from Lonza Group (Milan, Italy), and the monocyte/macrophage cell line U937 from the European Collection of Authenticated Cell Cultures (ECACC, Salisbury, UK, SOP ECACC/079). Mycoplasma contamination in cell cultures was regularly (monthly) tested by Real Time PCR (RT-PCR) MycoSPY Kit (Biontex, Munchen, Germany). Cells were grown at 37 °C in 95% air-5% CO_2_ in 100-mm^2^ cell culture dishes. Cells were fed every 3–4 days with the recommended medium EGM-2 MV BulletKit^TM^ Medium (Lonza Group) for HUVEC and HCAEC, and RPMI medium (Thermo Fisher Scientific, Waltham, MA, USA) supplemented with 10% fetal bovine serum, penicillin (100 U/mL) and streptomycin (100 mg/mL) for U937 cells.

### 2.2. Cell Treatments

HUVEC, HCAEC and U937 cells were grown under normal glucose (NG, 5.5 mM) and HG (10, 20 and 30 mM) conditions for different times (from 0 to 72 h), (1) in the presence or absence of ʟ-Carnosine (Car, 20 mM, Sigma-Aldrich, St. Louis, MO, USA), an endogenous dipeptide shown to protect against several oxidative-based diseases by trapping carbonyl compounds [22,23], including the toxic α-oxoaldehyde and glycolysis side product methyl glyoxal (MGO) [24,25], (2) with or without trans-resveratrol (Res, 10 µM, Sigma-Aldrich), a prototype inducer of the glyoxalase-1 (GLO1) enzyme that detoxifies α-oxoaldehydes, including MGO [26], or (3) in cells silenced or not for *HIF-1α*. Mannitol was used as osmotic control for HG effects on HIF-1α activity and proinflammatory activation of endothelial cells and macrophages (Appendix A). In U937 cells, the effects of HG were analysed upon LPS (10 ng/mL) stimulation. HUVEC and U937 cells were also incubated with MGO (0.2 mM, 48 h). Whole cell and nuclear proteins were extracted using the Whole Cell and Nuclear Extraction Kits (Abcam, Cambridge, UK), respectively.

### 2.3. mRNA and Nuclear Protein Levels of HIF-1α

The mRNA and nuclear protein levels of HIF-1α were assessed by RT-PCR using inventoried TaqMan gene expression assays (Applied Biosystems, Carlsbad, CA, USA, Table 1), and Western blot and immunofluorescence (IF), using specific anti-HIF-1α antibodies (Table 2). IF images were acquired using a 40x/N.A. 1.3 oil objective on a Zeiss Axiovert 200 M fluorescence microscope equipped with an Axiocam 503 color camera, controlled with ZEN (blue edition) software (Zeiss, Milan, Italy).

### 2.4. General Methods for mRNA and Protein Expression Analysis

RNA was isolated using the RNeasy Plus Mini Kit (Qiagen, Milan, Italy) and RNA reverse transcription was performed with High Capacity cDNA Reverse Transcription kit (Thermo Fisher Scientific). The StepOne ™ Real Time PCR System (Thermo Fisher Scientific) was used to quantify the relative gene expression levels and analyse the data. Gene expressions were calculated using the ΔΔCt method and were normalized to control (β-actin expression). Western blot experiments were performed as per manufacturer’s instructions. Briefly, protein samples were subjected to SDS-PAGE and transferred to PVDF membranes. Skin milk (5%) was used as blocking agent, while skin milk or BSA (5%) were used as incubation solutions depending on the antibodies and relative manufacturers’ instructions. Primary antibodies were incubated overnight at 4 °C and secondary antibodies at RT for 1 h. Blots were developed by enhanced chemiluminescence using Clarity or Clarity Max ECL substrates (Bio-Rad Laboratories, Milan, Italy). The chemiluminescent signal was detected and quantified by ChemiDoc XRS system (Bio-Rad Laboratories).

### 2.5. HIF-1α Activity

HIF-1α activity was assessed by luciferase assay using the Cignal HIF-1α Reporter (luc) Kit (Qiagen). Briefly, cells were transiently transfected with an inducible transcription factor-responsive firefly luciferase reporter using Attractene Transfection Reagent (Qiagen), according to the manufacturer’s instructions. The luminescence was measured using Dual-Luciferase Reporter Assay Kit (Promega, Milan, Italy).

### 2.6. HIF-1α Silencing

*HIF-1α* silencing (si-*HIF-1α*) was performed by Silencer^®^ Select & Validated small interfering RNA (Thermo Fisher Scientific) using the assay ID s6539 and a non-targeting negative control siRNA. Cells were transfected with Lipofectamine RNAiMAX Reagent (Invitrogen, Carlsbad, CA, USA).

### 2.7. Endothelial Cell and Macrophage Activation

The mRNA expression of vascular cell adhesion molecule 1 (*VCAM-1*) and monocyte chemoattractant protein-1/C-C motif chemokine ligand 2 (*MCP-1/CCL2*) in HUVEC, and *IL-1β* mRNA in U937 cells were assessed by RT-PCR (Table 1). The release of *IL-1β* and tumor necrosis factor (TNF)-α into the medium of U937 cells was assessed by using the Human *IL-1β*/IL-1F2 and TNF-α Quantikine ELISA kits from R&D Systems (Milan, Italy).

### 2.8. MGO-Mediated Post-Translational Modification and Hydroxylation Activity of PHD2

PHD2 mRNA (Table 1) and protein (Table 2) levels were assessed by RT-PCR and Western blot, respectively. To investigate the effect of MGO on PHD2 activity, proteasome activity was inhibited with 10 μM MG132 (Selleckchem Chemicals, Houston, TX, USA) for 6 h. Cytosolic and nuclear proteins were extracted using the Extraction Kits (Abcam) and analysed by Western blot for prolyl-hydroxylated (Pro-OH) HIF-1α and nuclear HIF-1α, respectively (Table 2). To explore MGO-mediated post-translational modification of PHD2, MGO protein adducts, PHD2 and mouse IgG immunoprecipitations (IP) were performed using the Pierce Crosslink Magnetic IP/Co-IP kit (Thermo Fischer Scientific, Waltham, MA, USA) according to manufacturer instruction (Table 2). Proteins were extracted in Pierce IP Lysis Buffer (Thermo Fischer Scientific). In brief, 25 µL of Protein A/G magnetic beads crosslinked to 5 µg of anti-PHD2/EGLN1, anti-MGO-modified proteins, or IgG2 Isotype control antibodies were incubated with 500 µg of cell lysate for 1 h at room temperature. Then, the beads were washed to remove non-bound material and a low pH elution buffer was used to dissociate bound antigen from the antibody-crosslinked beads. The IP proteins were analysed by Western blot (Table 2).

### 2.9. Expression Levels of Glycolytic Enzymes and Markers of Aerobic Metabolism

To investigate the HIF-1α-mediated increase in glycolysis and suppression of mitochondrial respiration and biogenesis, the protein levels of known targets of HIF-1 α, including the glycolytic enzymes hexokinase 2 (HK2), pyruvate kinase M2 (PKM2), lactate dehydrogenase A (LDHA) [12,14,16,27,28], and pyruvate dehydrogenase kinase 1 (PDK1), a serine/threonine kinase that inhibits mitochondrial respiration by inactivating pyruvate dehydrogenase [15], were evaluated by Western blot (Table 2). Aerobic metabolism was assessed by measuring mRNA expression of another HIF-1 α target, peroxisome proliferator-activated receptor gamma coactivator 1-α (*PGC-1α*) (Table 1), which is a positive regulator of mitochondrial biogenesis and function [29]. Mitochondrial function was also evaluated by IF analysis of mitochondrially encoded cytochrome c oxidase subunit 1 (MT-CO1) of respiratory Complex IV (Table 2), a key enzyme in aerobic metabolism. IF images were acquired as detailed above.

### 2.10. Monitoring of Mitochondrial Respiration and Glycolytic Flux

The shift from oxidative phosphorylation to aerobic glycolysis was analysed by evaluating cellular oxygen consumption and glycolysis in living cells using Oxygen consumption/Glycolysis Dual Assay Kit (Cayman Chemical, Ann Arbor, MI, USA). Briefly, the signal from a phosphorescent oxygen probe (MitoXpress^®^—Xtra) and the concentration of lactate released into the culture medium were evaluated and quantified by means of a multimode microplate reader (Varioskan Lux, Thermo Fischer Scientific).

### 2.11. Alternative Pathways of Glucose Metabolism and Superoxide Anion Inhibition

The flux of glucose through the polyol and hexosamine pathways was assessed using the D-Sorbitol Assay Colorimetric Kit (Abcam) and the Glutamine-Fructose-6-Phosphate Transaminase 1 (GFPT1) kit (Aviva Systems Biology, San Diego, CA, USA), respectively. Intracellular formation of AGEs was measured by the OxiSelect™ Advanced Glycation End-Product Competitive ELISA Kit (Cell Biolabs, San Diego, CA, USA). Polyethylene glycol-superoxide dismutase (PEG-SOD, 50 U/mL, Sigma-Aldrich), a cell permeable analogue of SOD, was used to quench superoxide.

### 2.12. Statistical Analysis

The numbers of biological (i.e., independent experiments) or technical (used to ensure the reliability of single values) replicates are reported in figure legends. Results are expressed as mean ± SEM and fold or % change vs. controls. For comparisons between two groups, unpaired Student’s t-tests with no assumption of equal variance were used. For comparisons of more than two groups, one-way ANOVA followed by the Tukey’s post-test for multiple comparisons or two-way ANOVA followed by the Bonferroni post-test were used, as appropriate. A *p*-value <0.05 was considered significant. All statistical tests were performed on raw data using GraphPad Prism version 5.00 for Windows (GraphPad Software, San Diego, CA, USA).

## 3. Results

### 3.1. HIF-1α Induction and Cell Activation by High Glucose

HG (20 mM) induced nuclear translocation of HIF-1α in both HUVEC (Figure 1A,B) and LPS-stimulated (10 ng/mL) U937 macrophage cell line (Figure 1C) at 48 h. Similar results to those in HUVEC were obtained in HCAEC (Figure 1D). In HUVEC, glucose induced HIF-1α nuclear translocation in a dose-dependent manner, ranging from 90% at 10 mM to ~9-fold at 30 mM after 48 h, and a weak increase in *HIF-1α* mRNA expression (Figure 1E), which reached significance only for the highest concentration (30 mM) at the longer time point of 72 h (Figure 1F). A weak and late effect on *HIF-1α* mRNA levels was also observed in LPS-stimulated U937 cells (Figure 1G), confirming that glucose-induced HIF-1α nuclear translocation can be accounted for by protein changes, not by transcriptional regulation. As assessed by luciferase gene reporter assay, HIF-1α activity was increased by 66% in HUVEC (Figure 2A) and >2-fold in LPS-stimulated U937 cells (Figure 2B) after 48 h incubation in HG.

HIF-1α nuclear translocation and activity were accompanied by upregulation of *VCAM-1* (2.9-fold, Figure 2C) and *MCP-1*/*CCL2*) (2-fold, Figure 2D) in HUVEC. In U937 cells, HIF-1α induction by HG was associated with increased mRNA levels of *IL-1β* (+50%, Figure 2E) and secretion of *IL-1β* (+84%, Figure 2F) and TNF-α (2-fold, Figure 2G). si-*HIF-1α* significantly reduced HG-induced proinflammatory cell activation, as it reduced the mRNA expressions of *VCAM-1* (−52%, Figure 2C) and *MCP-1* (−60%, Figure 2D) in HUVEC and reversed upregulation of *IL-1β* (Figure 2E) and secretion of *IL-1β* (Figure 2F) and TNF-α (Figure 2G) in U937 cells. Consistent with a critical role of HIF-1α signalling in macrophage activation (16–18), in addition to reversing the effect of HG, si-*HIF-1α* was even able to prevent LPS-dependent secretion of *IL-1β* (Figure 2F) and TNF-α (Figure 2G) in U937 cells.

### 3.2. α-Oxoaldehydes as Mediators of Glucose-Induced HIF-1α Induction

Car inhibited HG-induced nuclear translocation of HIF-1α in HUVEC (Figure 1A,B), U937 cells (Figure 1C), and HCAEC (Figure 1D). Car also inhibited HG-induced HIF-1α activity in HUVEC and U937 cells (Figure 3A,B) and significantly reduced the mRNA levels of *VCAM-1* (−48%) (Figure 2C) and *MCP-1* (−44%) (Figure 2D) compared with untreated HUVEC. Finally, Car normalized the *IL-1β* (Figure 2E) and *IL-1β* (Figure 2F) mRNA levels and TNF-α (Figure 2G) secretion into the culture medium of U937 cells. Similar effects in preventing HIF-1α nuclear translocation by HG were obtained by treating HUVEC (Figure 1A,B), U937 cells (Figure 1C), and HCAEC (Figure 1D) with Res. Consistently, treatment with MGO (200 µM) in NG (5.5 mM) conditions induced HIF-1α nuclear translocation in both HUVEC (Figure 3A) and U937 cells (Figure 3B) by ~6-fold and ~2-fold, respectively, an effect that was efficiently inhibited by the carbonyl trapping agent Car (Figure 3A,B).

To identify the mechanism involved in HIF-1α stabilization, we investigated whether MGO could affect mRNA and protein expression levels of PHD2, but obtained negative results (Figure 3C,D).

### 3.3. Inhibition of PHD2 Hydroxylation Activity by Post-Translational Glycation Mediated by MGO

We next investigated whether MGO could modify PHD2 post-translationally, thus affecting its hydroxylase activity. For this purpose, it was examined whether MGO-modified PHD2 could be detected in MGO-treated HUVEC. IP of MGO-treated cell extracts using a specific antibody for MGO modified proteins followed by PHD2 immunoblot analysis revealed a low but detectable basal level of MGO-modified PHD2 in HUVEC that was enhanced upon MGO treatment (Figure 4A).

IP of MGO-treated cell extracts using anti-PHD2 followed by immunoblot analysis of MGO protein adducts confirmed an enrichment of MGO-modified PHD2 in HUVEC cells treated with this glycolytic side product (Figure 4B).

To evaluate the magnitude of the effect of MGO on PHD2 activity, HUVEC cultured in NG were treated with MG132 to inhibit the proteasome activity and consequent degradation of (Pro-OH) HIF-1α (Figure 4C). Then, cytosolic extracts were tested by Western blot using an antibody specifically directed to (Pro-OH) HIF-1α, whereas the nuclear extracts were analysed using an anti-HIF-1α antibody that does not detect the Pro-OH form of HIF-1α. As expected, proteasome inhibition caused a build-up of (Pro-OH) HIF-1α in the cytosol of HUVEC. Conversely, proteasome inhibition in MGO-treated cells did not result in an accumulation of (Pro-OH) HIF-1α, indicating the suppression of PHD2 activity by MGO (Figure 4D). Western blot of nuclear extracts confirmed increased nuclear translocation of HIF-1α in cells treated with MGO (Figure 4E). Treatment with Car significantly restored PHD activity and reduced HIF-1α nuclear localization (Figure 4D,E) induced by MGO.

### 3.4. Modulation of HIF-1α Target Genes Related to Cellular Glucose Metabolism by MGO

In HUVEC cultured in NG and treated with MGO, protein levels of the glycolytic enzymes HK2 (Figure 5A), PKM2 (Figure 5B), and LDHA (Figure 5C) were increased by 75%, 32%, and 27%, respectively.

Protein levels of PDK1, an inhibitor of mitochondrial respiration, increased 2-fold (Figure 5D), whereas mRNA levels of *PGC-1α*, a positive regulator of mitochondrial biogenesis, decreased by 60% (Figure 5E) in MGO-treated cells. si-*HIF-1α* and Car treatment were both able to prevent the effects of MGO on glycolytic enzymes and mitochondrial function/biogenesis markers (Figure 5A–E). As compared with control cells, *PGC-1α* was even upregulated in si-*HIF-1α* cells (Figure 5E), a finding consistent with a feedback loop between *PGC-1α* and HIF-1α-signaling in controlling the rate of oxygen consumption and mitochondrial respiration [30]. In vivo IF-based evaluation of MT-CO1, a key enzyme in aerobic metabolism, confirmed the inhibitory effect of MGO on mitochondrial respiration, and the protection provided by Car (Figure 5F).

### 3.5. Contribution of HIF-1α-Related Cellular Energetic Changes to the Activation of the Alternative Pathological Pathways of Glucose Metabolism

In HUVEC, both HG and MGO reduced oxygen consumption (by 77% and 65%, respectively) at 60 min (Figure 6A,B) and increased lactate production (by 71% and 67%, respectively) at 48 h (Figure 6C,D).

Changes induced by HG and MGO in cellular glucose metabolism were significantly inhibited by Car (Figure 6A–D). In HG conditions, HIF-1α-related cellular energetic changes were paralleled by the activation of the glycolytic-side branches hexosamine pathway, as attested by the 70% increase in GFPT1 levels (Figure 6E), and polyol pathway, as demonstrated by the 2-fold increase in D-sorbitol (Figure 6F). As expected, AGE formation was also increased by HG treatment (+46%, Figure 6G). Both si-*HIF-1α* and Car treatment normalized the activity of the hexosamine and polyol pathways and prevented cellular accumulation of AGEs induced by HG (Figure 6E–G). Conversely, superoxide dismutation by PEG-SOD had no effect on the activity of the glycolytic side branches (Figure 6E–G).

## 4. Discussion

The accumulating evidence that mitochondrial superoxide generation is dispensable for the pathogenesis of vascular complications of diabetes [4,5,6,7]) compels us to search for new molecular triggers. This study provides a new mechanistic explanation, alternative to mitochondrial superoxide overproduction, for the initial changes in cellular metabolism of excess glucose and consequent biochemical abnormalities.

Here, we show that HIF-1α stabilization was an initial event in the cellular biochemical abnormalities induced by diabetic glucose concentrations. HIF-1α-dependent metabolic reprogramming induced by HG resembled the Warburg effect, as glucose flux was shifted from mitochondrial oxidation to glycolysis and lactate production. In turn, the accumulating glycolytic intermediates were channeled into alternative pathways that branch off from the glycolytic route and are known contributors to diabetic complications (i.e., polyol, hexosamine, and AGE pathways) [3,31]. The bioenergetic changes induced by HG were reversed by si-*HIF-1α*. Moreover, the carbonyl-trapping agent Car and the GLO1 inducer Res were able to inhibit HG-induced HIF-1α activation, suggesting a role for MGO, a toxic glucose metabolite produced from the spontaneous degradation of triose phosphates, the levels of which are increased when cells are exposed to glucose concentrations in the diabetic range [23,24]. Consistently, MGO was able to reproduce all the molecular, metabolic, and biochemical changes induced by HG, from HIF-1α stabilization and activity to endothelial cell and macrophage activation. Mechanistically, MGO induced HIF-1α nuclear translocation by interacting with PHD2, the main PHD isoenzyme responsible for HIF-1α degradation in normoxia [32], resulting in diminished enzyme activity.

These findings are consistent with the demonstration that, in mesangial and endothelial cells, HG activates HIF-1α and its target genes known to be involved in the development of micro- and macrovascular complications of diabetes [19,20]. It is also in line with a recent report showing that lactate production was increased in hyperglycemic myotubes in the absence of ROS generation [33]. However, it is at odds with the finding that, under hypoxic conditions, HIF-1α was inhibited by HG in renal proximal tubular cells [34,35], suggesting an impaired HIF-1α-dependent response to hypoxia in diabetes [36]. Actually, these apparently contrasting data may reflect differences in environmental conditions (normoxia versus hypoxia) and possibly cell type [36], suggesting that HIF-1α may serve as the initial trigger of glucotoxicity under normoxia by inducing a Warburg effect, but also mediate adaptation to tissue hypoxia, which is known to play a key role in diabetes-associated end-organ damage [37,38,39]. Therefore, while detrimental in an early phase, HIF-1α activity might exert a favorable effect in advanced disease by contrasting hypoxia. Results of in vivo studies in experimental animal models might also reflect this dual role of HIF-1α in diabetes-related vascular damage. On the one hand, HIF-1α activation was shown to promote atherosclerosis initiation and progression [40,41] as well as hypoxia-induced renal fibrogenesis [42,43,44] by stimulating epithelial-to-mesenchymal transition [45] and a2(I) collagen expression through interactions with Smad3 [46]. On the other hand, the *HIF-1A* Pro582Ser polymorphism, which confers less sensitivity to the inhibitory effect of glucose during a hypoxic challenge, was shown to be protective toward development of advanced diabetic nephropathy [47] and retinopathy [48]. In addition, inhibitors of PHDs provided protection against atherosclerosis in LDL receptor-deficient mice [49], and improved post-ischemia myocardial dysfunction [38] and renal metabolic abnormalities [50,51] in experimental models of diabetes. However, it is unclear whether renoprotection was mechanistically linked to the activation of HIF-1α or, rather, of the HIF-2α isoform, which is a key regulator of renal erythropoiesis [21,52] and a potential new target for the treatment of anemia in chronic kidney disease [53]. Furthermore, PHD2 overexpression attenuated the profibrotic effect of albumin in cultured renal tubular cells [54], whereas PHD2 ablation promoted renal vascular remodeling and fibrosis in mice [55], suggesting a detrimental role of unrestricted HIF-1α activity.

Known target genes of HIF-1α drove upregulated glycolysis and repressed mitochondrial respiration. In fact, HIF-1α induction was associated with an overexpression of the negative regulator of mitochondrial respiration PDK1 and the glycolytic enzymes HK, PKM2, LDH, whereas the mRNA levels of *PGC-1α*, an inducer of mitochondrial biogenesis and function, were reduced. At first view, our findings seem to be at odds with recent proteomic studies in individuals with very long diabetes duration (≥50 years), which indicated that glycolytic enzymes, particularly PKM2, were elevated in glomeruli and plasma of patients without diabetic nephropathy compared with patients with this complication, thus suggesting a protective role for PKM2 by increasing glucose flux through glycolysis [56,57]. It is known that PKM2 is modulated by HIF-1α [28] and cooperates with it in the regulation of aerobic glycolysis [58,59,60]. However, at variance with the constitutively active (i.e., tetrameric) PKM1 isoform, PKM2 is mainly present in a dimeric form, which is much less active in catalyzing the last step within glycolysis, thus favoring accumulation of upstream glycolytic metabolites, as occurs in cancer cells [61]. Conversely, in its dimeric form, PKM2 displays a unique gene regulatory function, complementary to that of HIF-1α, in the establishment of the Warburg effect [62]. Therefore, it may be speculated that, in diabetic patients without nephropathy, undetermined allosteric effectors stabilize PKM2 tetrameric assembly, thus favoring the glycolytic over the transcriptional regulator activity and the re-activation of mitochondrial oxidative metabolism. This interpretation is consistent with previous findings in preclinical models of diabetic nephropathy by several investigators [63,64], including Qi et al. [56], showing that pharmacological activation of PKM2 was associated with recovered mitochondrial function and biogenesis, suppressed HIF-1α and lactate accumulation, blunted elevation in toxic glucose metabolites, and reduced renal inflammation and fibrosis.

Diabetes is also considered a “chemical” disorder, as its long-term sequelae are the result of broad-based alterations in the chemical structure of biological molecules [65]. The hypothesis of HIF-1α induction as an early trigger of glucotoxicity combines metabolic and chemical mechanisms: higher intracellular glucose flux inevitably leads to increased MGO production, non-enzymatic post-translational glycation and PHD2 inhibition, with consequent HIF-1α-dependent reprogramming of cellular glucose metabolism (i.e., Warburg effect), persistent elevation of glycolytic intermediates, and feeding of the biochemical (polyol, hexosamine, and AGE) pathways leading to cell injury. This is consistent with several studies showing that Car, or its derivatives, protected diabetic mice from vascular complications [66,67,68] and that protection was associated with hindered HIF-1α upregulation in the vasculature of type 1 [67] and liver of type 2 [69] diabetic mice. In addition, it is in keeping with the findings that *Glo1* knockdown in nondiabetic mice mimics [70], whereas *Glo1* overexpression in diabetic animals improves [71] microvascular pathology. The implication of this mechanistic hypothesis is that HIF-1α mediated metabolic reprogramming is sufficient, whereas mitochondrial superoxide production is dispensable, for inducing vascular cell activation, inflammation and AGE accumulation, three well recognized contributors to vascular dysfunction and pathology in diabetes [2,3,31].

Although our results argue against mitochondrial superoxide production as the initial trigger, they do not exclude the detrimental role of oxidative stress in vascular complications of diabetes, as most of the pathways that branch off from glycolysis produce reactive oxygen species [2,72,73], including mitochondrial superoxide generated in response to AGE binding to their receptor [74,75]. The concept that oxidative stress is a downstream event, not the initiator of the cellular events triggered by glucose, may explain the failure of clinical trials to prove a clear benefit of antioxidants in diabetic patients [76], though other mechanisms may be responsible for these discouraging results [77]. Finally, considering that not all reactive carbonyls require increased oxidative stress for their formation in vivo [65], carbonyl stress, intended as an increased production of glycolytic, non-oxidatively derived carbonyls such as MGO, is a more likely candidate than oxidative stress as initial mediator of glucotoxicity. This implies that interventions using carbonyl sequestering agents and/or inducers of the glyoxalase pathway may be more effective than antioxidants in preventing vascular complications [66,67,68,69,70,71].

The main limitation of this study is the lack of in vivo data. However, direct targeting of HIF-1α or PHD2 in diabetic animal models cannot provide further comprehension of their role in the instigation of the initial cellular biochemical abnormalities induced by hyperglycemia, which is the aim of this study. Another limitation may be the lack of specificity of Car and Res in protecting from glucose-induced changes of HIF-1α activity, as these compounds have other pleiotropic effects beyond their MGO-sequestering and GLO-1 inducer activities. In fact, both Car and Res have been shown to exert indirect antioxidant effects to upregulate endogenous antioxidants and other cell stress response genes [78,79,80,81]. However, the efficacy of Car to prevent HIF-α changes induced by MGO, including the reduced prolyl hydroxylation by PHD2, can only be attributed to its carbonyl trapping activity, thus supporting the concept that the protective effects exerted by both compounds in HG conditions were mediated by their ability to prevent glycolysis-induced MGO accumulation.

## 5. Conclusions

In conclusion, this study identifies HIF-1α and related cellular energetic changes (i.e., Warburg effect) as early and essential components of a metabolic and chemical process that mediates the injurious effect of hyperglycemia, leading to cell activation and inflammation.

## Figures and Tables

**Figure 1 biomedicines-09-01139-f001:**
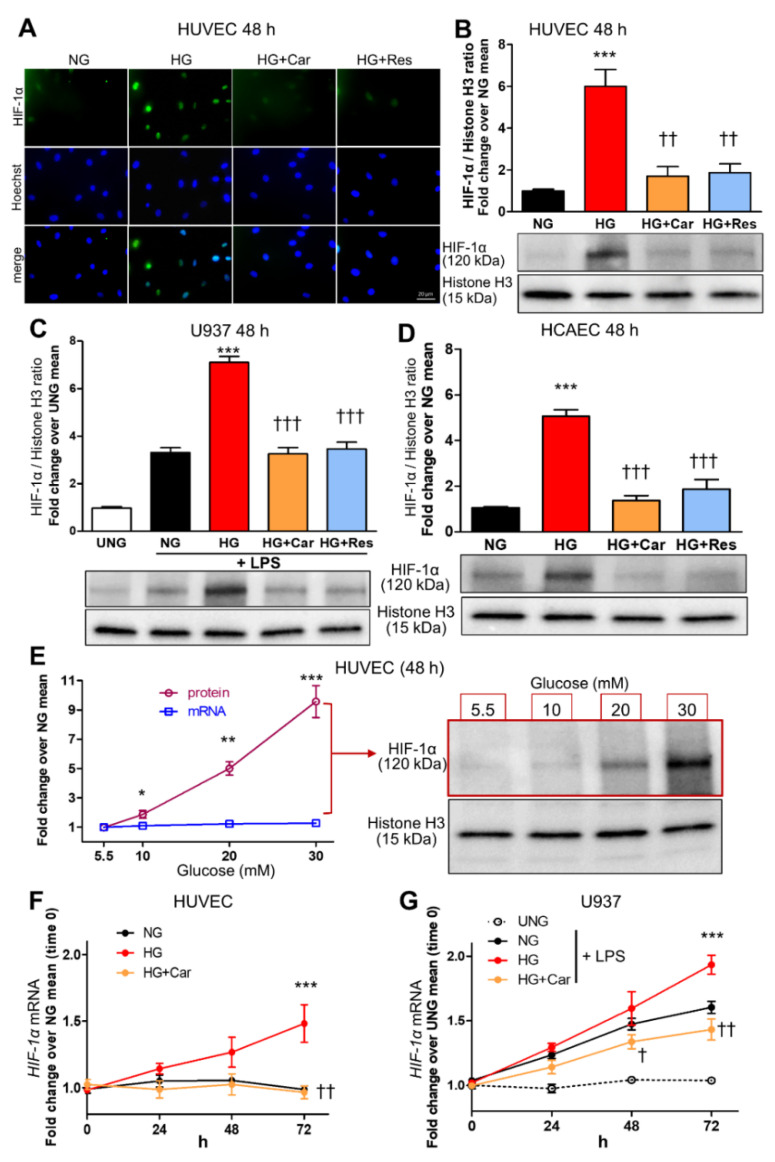
HG increases HIF-1α nuclear translocation in endothelial cells and LPS-stimulated macrophages. Representative IF ((**A**), scale bar 20 µm) and Western blot (**B**) for nuclear HIF-1α in HUVEC exposed to HG (20 mM) vs. NG (5.5 mM) for 48 h, with or without the carbonyl trapping agent Car (20 mM) or the glyoxalase inducer Res (10 µM), and relative band densitometry analysis from three separate experiments. Western blot analysis for nuclear HIF-1α in U937 cells stimulated (or not, UNG = untreated NG) with LPS (10 ng/mL) (**C**), and HCAEC (**D**), after 48 h incubation with HG vs. NG with or without Car or Res, and relative band densitometry analysis from three separate experiments. Dose response curve (**E**) of nuclear HIF-1α protein levels (*n* = 3 separate experiments per condition, black line) and mRNA HIF-1α expression (*n* = 5 wells in duplicate per condition, blue line) in HUVEC exposed to varying concentrations of glucose ranging from 5,5 mM to 30 mM for 48 h, and representative Western blot image (right side of panel (**E**)). Time course of HIF-1α mRNA expression in HUVEC (**F**) and U937 cells stimulated (or not, UNG) with LPS (**G**), exposed to HG vs. NG for different times ranging from 0 to 72 h, with or without Car; *n* = 5 wells in duplicate per time point per condition. Bars represent mean ± SEM. Post hoc multiple comparison: *** *p* < 0.001, ** *p* < 0.01 or * *p* < 0.05 vs. NG; ††† *p* < 0.001, †† *p* < 0.01 or † *p* < 0.05 vs. HG.

**Figure 2 biomedicines-09-01139-f002:**
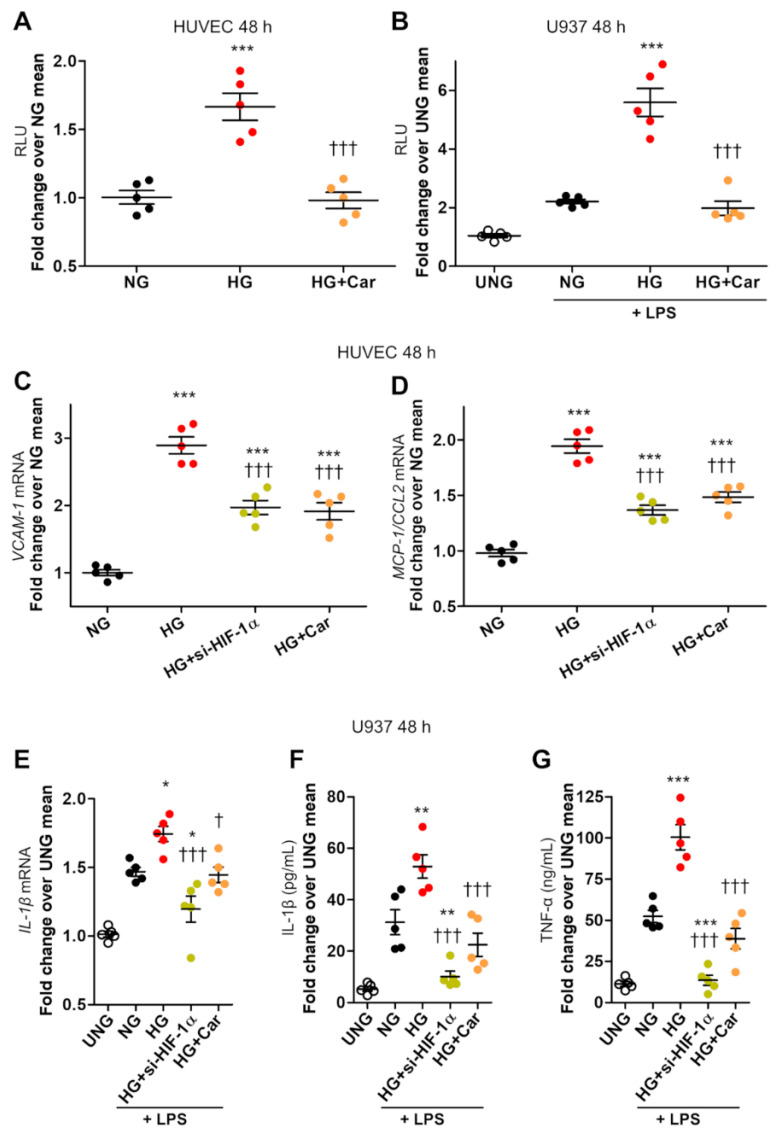
HG-induced HIF-1α activity is associated with proinflammatory activation in endothelial cells and LPS-stimulated macrophages. HIF-1 activity, as assessed by dual-luciferase gene reporter assay, in HUVEC (**A**), and in U937 macrophages stimulated (or not, UNG = untreated normal glucose) with LPS (10 ng/mL) (**B**), after 48 h incubation with HG (20 mM) vs. NG (5.5 mM), with or without Car (20 mM); *n* = 5 wells in duplicate per condition. *VCAM-1* (**C**) and *MCP-1*/*CCL2* (**D**) mRNA levels in HUVEC, and *IL-1β* mRNA levels in U937 macrophages stimulated (or not, UNG) with LPS (**E**), exposed to HG vs. NG for 48 h, silenced for *HIF-1α* (si-*HIF-1α*), or treated with Car; *n* = 5 wells in duplicate per condition. *IL-1β* (**F**) and TNF-α (**G**) protein levels in the culture medium of U937 cells stimulated (or not, UNG) with LPS after 48 h incubation with HG vs. NG, silenced for *HIF-1α*, or treated with Car; *n* = 5 wells in duplicate per condition. Each dot represents the mean of two individual technical replicate and bars represent mean ± SEM. Post hoc multiple comparison: *** *p* < 0.001, ** *p* < 0.01 or * *p* < 0.05 vs. NG; ††† *p* < 0.001 or † *p* < 0.05 vs. HG.

**Figure 3 biomedicines-09-01139-f003:**
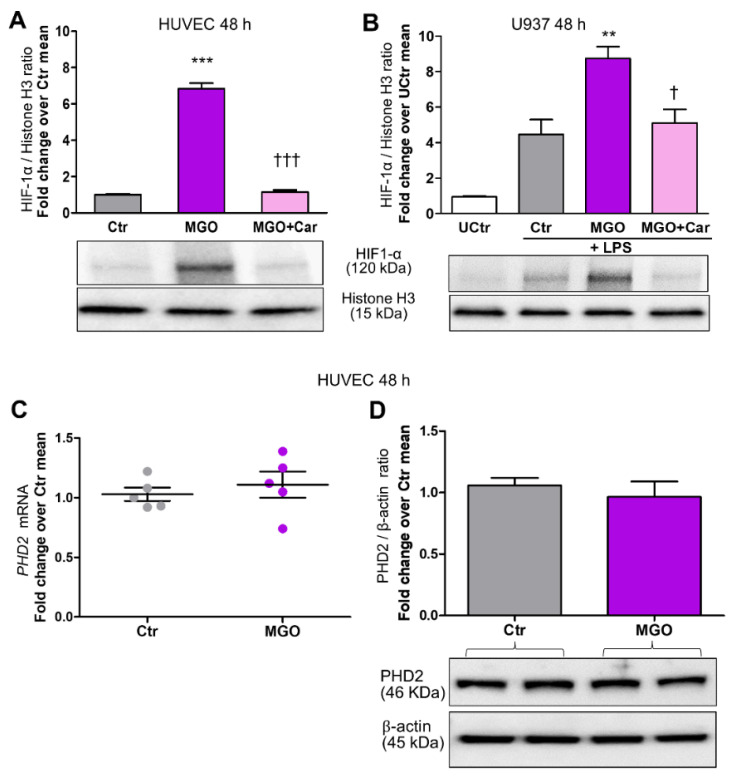
The glycolytic side-product MGO increases HIF-1α nuclear translocation in endothelial cells and LPS-stimulated macrophages but does not affect mRNA and protein levels of PHD2. Western blot analysis for HIF-1α in nuclear extracts from HUVEC (**A**) and U937 cells stimulated (or not, UCtr = untreated control) with LPS (10 ng/mL) (**B**), treated or untreated (Ctr) with the reactive dicarbonyl compound MGO (200 µM) for 48 h, in the presence or absence of the carbonyl trapping agent Car (20 mM), and relative band densitometry analysis from three separate experiments. *PHD2* mRNA ((**C**), *n* = 5) and protein levels in total extracts (**D**), from HUVEC treated or untreated (Ctr) with MGO (200 µM) for 48 h, and relative band densitometry analysis from three separate experiments. Each dot in (**C**) represents the mean of two individual technical replicate. Bars represent mean ± SEM. Post hoc multiple comparison: *** *p* < 0.001 or ** *p* < 0.01 vs. Ctr; ††† *p* < 0.001 or † *p* < 0.05 vs. MGO.

**Figure 4 biomedicines-09-01139-f004:**
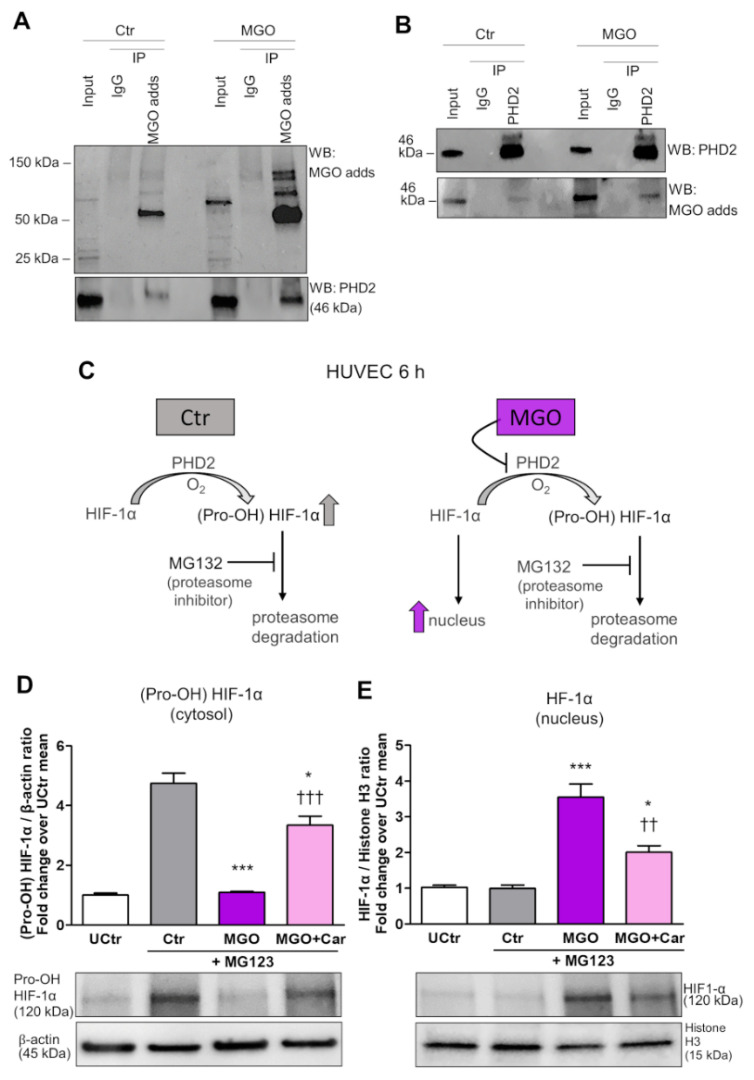
The glycolytic side-product MGO activates nuclear translocation of HIF-1α by inducing post-translational glycation and inhibition of PHD2 activity. IP of MGO-protein adducts (MGO adds) (**A**) or PHD2 (**B**) on HUVEC treated (MGO) or not (Ctr) with 200 µM MGO for 6 h, using a specific antibody for MGO modified proteins (**A**) or a specific anti-PHD2 antibody (**B**), respectively. Mouse IgG were used as control. Total cell lysates (Input) and immunoprecipitates were immunoblotted for PHD2 and MGO protein adducts (**A**,**B**). In the presence of oxygen, HIF-1α is rapidly hydroxylated by PHD2 to generate (Pro-OH) HIF-1α, which is degraded through the ubiquitin-proteasome pathway; proteasome inhibition with MG132 led to Pro-OH HIF-1α increase, and adding MGO prevented (Pro-OH) HIF-1α increase and led to HIF-1α nuclear translocation (**C**). Western blot analysis for (Pro-OH) HIF-1α in total extracts (**D**), and non-hydroxylated HIF-1α in nuclear extracts (**E**) from HUVEC, treated or untreated (UCtr) with the proteasome inhibitor MG132 (10µM) for 6 h, with or without MGO, in the presence or absence of Car. Bars represent mean ± SEM. Post hoc multiple comparison: *** *p* < 0.001 or * *p* < 0.05 vs. Ctr; ††† *p* < 0.001 or †† *p* < 0.01 vs. MGO.

**Figure 5 biomedicines-09-01139-f005:**
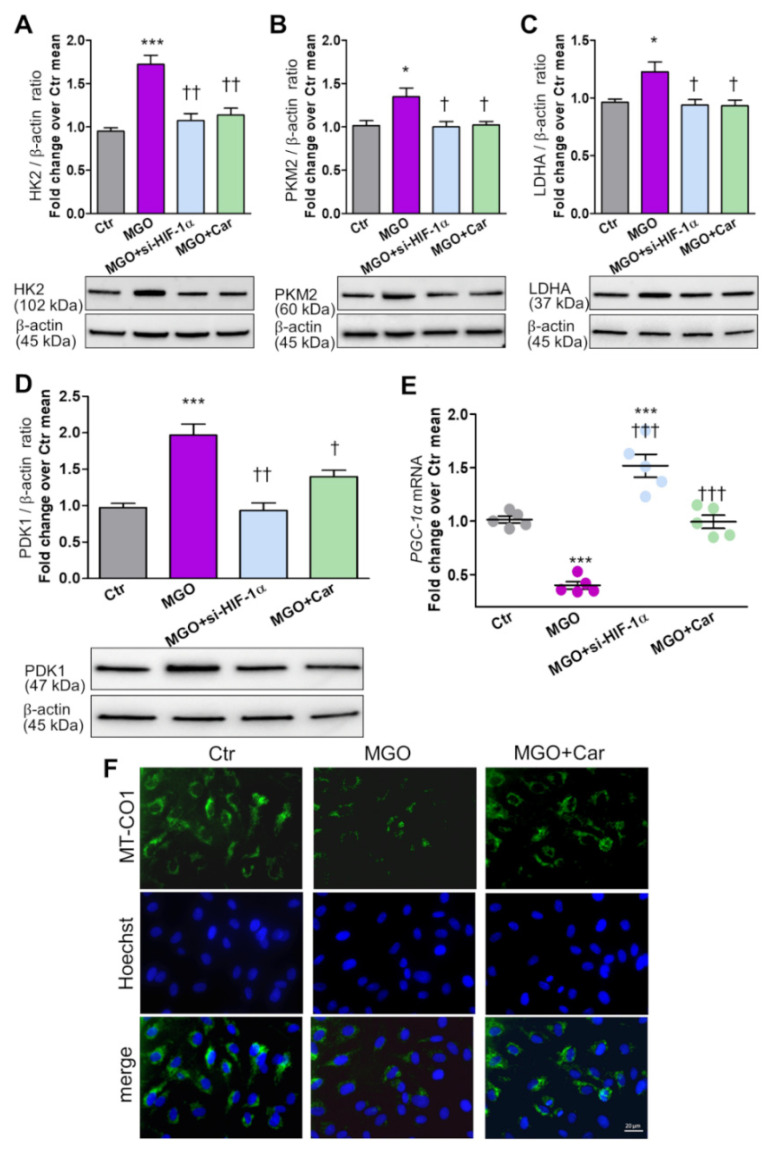
The glycolytic side-product MGO modulates HIF-1α target genes related to cellular glucose metabolism. Western blot analysis of the glycolytic enzymes HK2 (**A**), PKM2 (**B**) and LDHA (**C**), and of the inhibitor of pyruvate dehydrogenase activity PDK1 (**D**) in total cell extracts from HUVEC treated (or untreated, Ctr = control) with the reactive dicarbonyl compound MGO (200 µM) for 48 h, silenced for *HIF-1α* (si-*HIF-1α*), or in the presence of the carbonyl trapping agent Car (20 mM), and relative band densitometry analysis from three separate experiments. mRNA levels of the mitochondrial biogenesis marker *PGC-1*α (**E**) in HUVEC exposed to MGO for 48 h, silenced for *HIF-1α*, or treated with Car; each dot represents the mean of two technical replicates of 5 wells per condition. Representative IF (**F**, scale bar 20 µm) for MT-CO1 in HUVEC treated or untreated (Ctr) with MGO for 48 h, in the presence or absence of Car. Bars represent mean ± SEM. Post hoc multiple comparison: *** *p* < 0.001 or * *p* < 0.05 vs. Ctr; ††† *p* < 0.001, †† *p* < 0.01 or † *p* < 0.05 vs. MGO.

**Figure 6 biomedicines-09-01139-f006:**
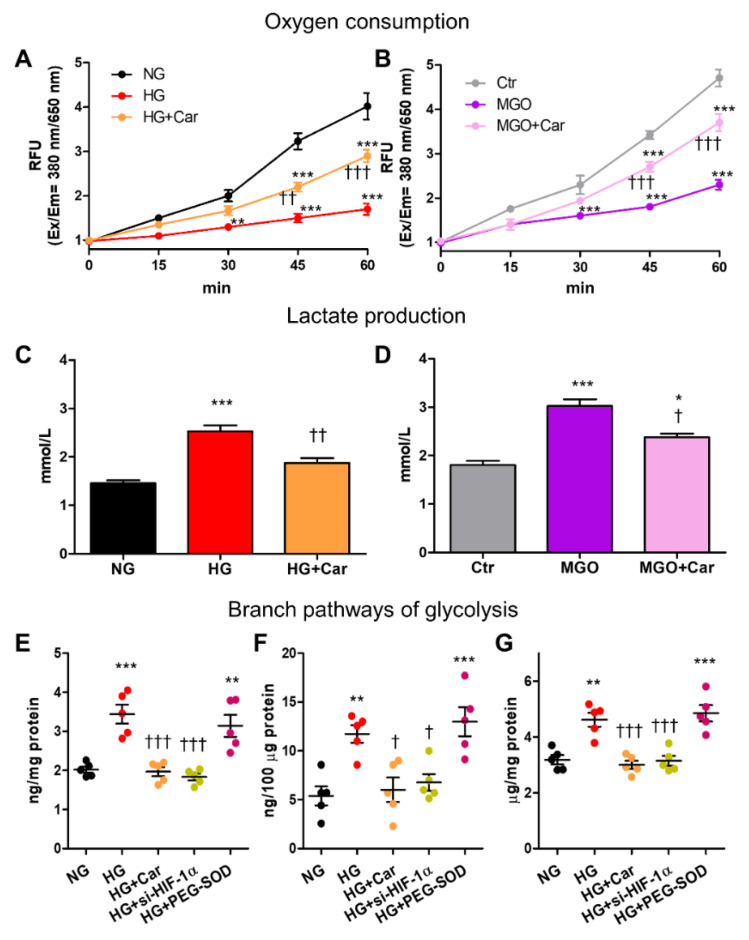
HG and the glycolytic side product MGO induce cellular energetic changes that resemble the Warburg effect and are associated with the activation of the alternative pathological pathways of glucose metabolism in HG conditions. Time course of oxygen consumption (**A**,**B**) and lactate production at 48 h (**C**,**D**) by HUVEC exposed to HG (20 mM) vs. NG (5.5 mM) (**A**,**C**), or treated with MGO (200 µM) vs. untreated cells (Ctr) (**B**,**D**), with or without the carbonyl trapping agent Car (20 mM); n = 3 separate experiments in duplicate per condition. Hexosamine (**E**), polyol (**F**) and AGE (**G**) pathways activation in HUVEC exposed to HG vs. NG for 48 h, with or without Car, as determined by measuring the levels of GFPT1 (**E**), D-sorbitol (**F**), and AGEs (**G**); each dot represents the mean of two technical replicates of 5 wells per condition. Bars represent mean ± SEM. Post hoc multiple comparison: *** *p* < 0.001, ** *p* < 0.01 or * *p* < 0.05 vs. NG or Ctr (as appropriate); ††† *p* < 0.001, †† *p* < 0.01 or † *p* < 0.05 vs. HG or MGO (as appropriate).

**Table 1 biomedicines-09-01139-t001:** TaqMan Gene Expression assays.

Target	Assay
*HIF-1α*	Hs00153153_m1 (#4331182)
*VCAM-1*	Hs01003372_m1 (#4331182)
*MCP-1*/*CCL2*	Hs00234140_m1 (#4331182)
*IL-1β*	Hs0155410_m1 (#4331182)
*PGC-1α*	Hs00173304_m1 (#4331182)
*PHD2*/*EGLN1Β-actin*	Hs00254392_m1 (#4331182)
*Β-actin*	Hs99999903_m1 (#4331182)

*HIF-1α* = hypoxia inducible factor 1α; *VCAM-1* = vascular cell adhesion molecule 1; *MCP-1*/*CCL2* = monocyte chemoattractant protein 1/C-C motif chemokine ligand 2; *IL-1β* = interleukin 1β; *PGC-1α* = peroxisome proliferator-activated receptor gamma coactivator 1α; *PHD2*/*EGLN1* = prolyl hydroxylase domain-containing protein 2/egl-9 family hypoxia inducible factor 1.

**Table 2 biomedicines-09-01139-t002:** Antibodies used in Western blot and IF studies.

Target	Antibody	Catalog Nr.	Supplier
**Primary**			
HIF-1α (WB)	Rabbit polyclonal	#3716	Cell Signaling Technology, Leiden, The Netherlands
HIF-1α (IF)	Rabbit monoclonal	ab51608	Abcam, Cambridge, UK
Pro-OH HIF-1α (WB)	Rabbit monoclonal	3434	Cell Signaling Technology, Danvers, MA, USA
PHD2/EGLN1 (IP-WB)	Rabbit polyclonal	NB100-137	Novus Biologicals, Centennials, CO, USA
HK2 (WB)	Mouse monoclonal	ab104836	Abcam, Cambridge, UK
PKM2 (WB)	Rabbit polyclonal	#3198	Cell Signaling Technology, Danvers, MA, USA
LDHA (WB)	Rabbit polyclonal	19987-1-AP	Proteintech, Manchester, UK
PDK1 (WB)	Rabbit monoclonal	#3820	Cell Signaling Technology, Danvers, MA, USA
MT-CO1 (IF)	Mouse monoclonal conjugated to Alexa Fluor^®^ 488	Ab154477	Abcam, Cambridge, UK
MGO (IP-WB)	Mouse monoclonal	NBP-2 59368	Novus Biologicals, Centennials, CO, USA
IgG2 Isotype control (IP)	Mouse monoclonal	ab18415	Abcam, Cambridge, UK
IgG Isotype control (IP)	Rabbit monoclonal	Ab172730	Abcam, Cambridge, UK
*Β-actin*	Mouse monoclonal	A5441	Sigma Aldrich, St. Louis, MO, USA
Histone H3 (WB)	Rabbit polyclonal	ab1791	Abcam, Cambridge, UK
**Secondary**			
HIF-1α (WB), Pro-OH HIF-1α, PHD2/EGLN1, PKM2, LDHA, PDK1	HRP-conjugated goat anti-rabbit	P0448	Agilent/Dako, Santa Clara, CA, USA
B-actin, HK2, MGO	HRP-conjugated goat anti-mouse	P0447	Agilent/Dako, Santa Clara, CA, USA
HIF-1α (IF)	Alexa Fluor^®^ Plus 488 goat anti-rabbit IgG	A-32731	Thermo Fisher Scientific, Waltham, MA, USA

HIF-1α = hypoxia inducible factor 1α; WB = Western blot; IF = immunofluorescence; Pro-OH HIF-1α = proline-hydroxylated hypoxia-inducible factor 1α; PHD2/EGLN1 = prolyl hydroxylase domain-containing protein 2/egl-9 family hypoxia inducible factor 1; HK2 = hexokinase 2; PKM2 = pyruvate kinase isozymes M2; LDHA = lactate dehydrogenase A; PDK1 = pyruvate dehydrogenase kinase 1; MT-CO1 = mitochondrially encoded cytochrome C oxidase I; MGO = methylglyoxal; IP = immunoprecipitation.

## Data Availability

The datasets used and/or analysed during the current study are available from the corresponding author on reasonable request.

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
