# Peer review of "Normalizing HIF-1α Signaling Improves Cellular Glucose Metabolism and Blocks the Pathological Pathways of Hyperglycemic Damage"

_biomedicines, 2021, doi:10.3390/biomedicines9091139_

Round 1

Reviewer 1 Report

The authors made a huge effort to perform extensive experiments to check for the contribution of HIF-1alpha in the development of diabetic complications.

I am highly impressed by the style of writing, the variety of methods used in this study, the high amount of time and effort which was spent to write this paper. 

I have some things to be considered before acceptance for publication : 

1) some editorial errors need correction (for example - 2.6 or 3.1 section (the latter has some crossed out phrases - Fig 3A and 3B)). 

2) I did not find the information about normalization of real time data? What genes or pair of genes were used ? Did you check for their suitability before experiments (lack of variation in their expression level under HG?) in each type of cells used in this paper? The same question about western blot - did you check for b-actin suitability before experiments? 

I admire how comprehensively authors discussed obtained results! The conclusions provide an answer for tested hypothesis. 

Author Response

Reviewer #1

The authors made a huge effort to perform extensive experiments to check for the contribution of HIF-1alpha in the development of diabetic complications. I am highly impressed by the style of writing, the variety of methods used in this study, the high amount of time and effort which was spent to write this paper.

We thank the Reviewer for the positive comment to our work

I have some things to be considered before acceptance for publication:

  1. Some editorial errors need correction (for example - 2.6 or 3.1 section (the latter has some crossed out phrases - Fig 3A and 3B)).

Thanks for reporting these typos, which have been corrected (see tracked changes in “2.6” and “3.2” sections)

  1. I did not find the information about normalization of real time data? What genes or pair of genes were used? Did you check for their suitability before experiments (lack of variation in their expression level under HG?) in each type of cells used in this paper? The same question about western blot - did you check for β-actin suitability before experiments?

We apologize to the Reviewer for omitting this important information. We used the β -actin gene which, in our (Menini et al, Diabetes 2006) and other’s experience (Tseng et al, Front Immunol 2017 ), is a reliable reference gene in similar experimental conditions.  In fact, the β-actin gene expression is not affected by high glucose (Bakhashab et al, Investigation, 2014). We have now added the TaqMan assay ID in Table 1.

I admire how comprehensively authors discussed obtained results! The conclusions provide an answer for tested hypothesis.

We thank the reviewer again for the appreciation of our work.

Reviewer 2 Report

The study by Carla Iacobini et al investigated an alternative pathway of pathological pathways under high glucose conditions in three different cell models. The study is original and the manuscript well written. Some aspects need to be improved before publication can be supported.The methods are not adequately described, the scientific soundness can therefore not be jduged at this point.

·       Abstract: provide the Hypothesis

·       L.35: specify: what do you mean by “defective response”

·       The materials and methods section does not provide sufficient information to judge the quality of the data, it needs to be thoroughly revised:

o   Tables 1&2 should be moved to the respective method section

o   Cell culture: how often is a regularly for the mycoplasma test? One per year/month/week?

o   Cell numbers and plate formats used for the different assays is constantly missing

o   The PCR method is almost lacking all required information. Please refer to the MIQE guidelines and provide at least information regarding the devices, reagents, controls used. How was the RNA isolated? How was the gDNA removed? Did the primers have an exon span? Was the product verified by sequencing? Was there a RT+ and RT- control used? How was RNA transcribed into cDNA? Which reference genes were used and how was the stability verified? Mastermix? How was the data analyzed?

o   Same is true for the Western Blots: all required information is lacking: original data need to be shown, incubation times with antibodies, blocking, how was detected

o   How were the different methods for protein determination chosen? When WB, when IF, ELISA?

o   Data for the specificity of the antibodies (blocking peptides) need to be presented at least in the supplemental material?

o   How were the loading controls selected?

o   LOD for the ELISAs need to be presented

·       Mannitol controls for osmotic effects are mentioned- they should be included in the figures

·       The scaling for the y-axis of figures showing the same type of data need to be adjusted to the same level since they are misleading regarding the effect size?

·       Why are the data presented with SEM instead of SD (beside the obvious effect of making the graphs look nicer?)

·       In general, an overview figure with the conclusion / identified pathway would make it more easy to follow the main story

·       There is no TOC graphics?

·       Figure 4 A & B & C a rather small and hard to read

Author Response

Reviewer #2

The study by Carla Iacobini et al investigated an alternative pathway of pathological pathways under high glucose conditions in three different cell models. The study is original and the manuscript well written. Some aspects need to be improved before publication can be supported.The methods are not adequately described, the scientific soundness can therefore not be jduged at this point.

We thank the Reviewer for his/her comments and criticisms, which we have addressed below in a point-by-point discussion. We believe we have satisfied all the reasonable requests, while discussing in this rebuttal those that we found inappropriate.

Abstract: provide the Hypothesis.

As requested by the Reviewer, the Abstract has been modified to add the hypothesis tested while respecting the maximum word limit (i.e., 200)

L.35: specify: what do you mean by “defective response”.

We have now clarified this point: “…..defective response to lipopolysaccharide (LPS) (i.e., loss of inflammatory capacity)”.

The materials and methods section does not provide sufficient information to judge the quality of the data, it needs to be thoroughly revised:

To address this criticism, we added a new subsection titled “2.4. General methods for mRNA and protein expression analysis” in the revised version of the manuscript.

 Tables 1&2 should be moved to the respective method section.

As requested, we moved Tables 1 and 2 to the method section “2.3. mRNA and nuclear protein levels of HIF-1α”.

- Cell culture: how often is a regularly for the mycoplasma test? One per year/month/week?

We usually check for mycoplasma contamination every month. We added this information in the main text.

- Cell numbers and plate formats used for the different assays is constantly missing.

Honestly, we do not believe that this is an essential information to be included in the manuscript. In fact, it cannot be found in any published paper, if anything, for space constraints. Therefore, we here provide some information regarding these aspects. We used different plate formats starting from 96 wells plates (for HIF-1α activity assay) up to 100 mm diameter dishes (for nuclear protein extraction). The cell number seeded was obviously proportional to the plate surface. For example, in 35 mm dishes we seeded 8x105 cells for U937 and 1.7x105 cells for HUVEC and HCAEC.

- The PCR method is almost lacking all required information. Please refer to the MIQE guidelines and provide at least information regarding the devices, reagents, controls used. How was the RNA isolated? How was the gDNA removed? Did the primers have an exon span? Was the product verified by sequencing? Was there a RT+ and RT- control used? How was RNA transcribed into cDNA? Which reference genes were used and how was the stability verified? Mastermix? How was the data analyzed?

We apologize with the Reviewer for omitting the information about reference gene. We used the β-actin gene that, in our experience (Menini et al, Diabetes 2006) and other’s experience (Tseng et al, Front Immunol 2017), is a reliable reference gene under these experimental conditions. In fact, the β-actin gene expression is not affected by high glucose (Bakhashab et al, Investigation, 2014). We have now added the TaqMan assay ID β-actin in Table 1 and most of the information requested by the Reviewer, including: Device (StepOne Real Time PCR System, Thermo Fisher Scientific); RNA isolation (RNeasy Plus Mini Kit, Qiagen #74134); and RNA reverse transcription (High Capacity cDNA Reverse Transcription kit, Thermo Fisher Scientific #4368814). About the other information requested, again we do not believe that these details should be included in the manuscript, as they can be easily found in the datasheet of each product. Therefore, we are happy to provide the Reviewer with the requested details in this rebuttal. About gDNA removal, the kit used for RNA isolation provides gDNA eliminator columns. Moreover, we used TaqMan assays in which the probes span an exon junction and do not detect the genomic DNA. We used predesigned TaqMan gene expression assays (Applied Biosystems), which are ready for use validated assays, with no needs for positive or negative controls. TaqMan gene expression assays are the gold standard for specificity, sensitivity, and reproducibility. The amplification reaction was performed with TaqManTM Gene Expression Master Mix. Finally, the data were analysed by the instrument software.

- Same is true for the Western Blots: all required information is lacking: original data need to be shown, incubation times with antibodies, blocking, how was detected

In the revised manuscript, we have now stated that “Western blot experiments were performed as per manufacturer's instructions. Briefly, protein samples were subjected to SDS-PAGE and transferred to PVDF membranes. Skin milk (5%) was used as blocking agent, while skin milk or BSA (5%) were used as incubation solutions depending on the antibodies and relative manufacturers’ instructions. Primary antibodies were incubated overnight at 4°C and secondary antibodies at RT for 1 hour. Blots were developed by enhanced chemiluminescence using Clarity or Clarity Max ECL substrates (Bio-Rad Laboratories, Milan, Italy). The chemiluminescent signal was detected and quantified by ChemiDoc XRS system (Bio-Rad Laboratories).” Regarding original data, the full unedited gel images have already been submitted; we believe they can be downloaded from the website or requested to the Editor.

- How were the different methods for protein determination chosen? When WB, when IF, ELISA?

Again, we do not believe that this is an essential information to be included in the manuscript. Western blot was used to investigate nuclear translocation of HIF-1α and measure cellular protein levels when a validated ELISA kit was not available; IF was used to show protein localization; ELISA was used to obtain quantitative determinations of cellular and secreted protein levels when a validated commercial kit was available.

- Data for the specificity of the antibodies (blocking peptides) need to be presented at least in the supplemental material?

We have used validated antibodies, not home-made antibodies. The antibodies datasheets provide information about target specificity and positive controls. Does the Reviewer think that we should had checked the antibody specificity further?

- How were the loading controls selected?

As for RT-PCR data, we used β-actin as loading control for total and cytosol protein extracts. In our previous experience (Menini et al, Diabetes 2006), β-actin is a reliable reference protein in experimental conditions in which glucose concentrations are altered (Bakhashab et al, Investigation, 2014). The same is true for Histone H3 gene, which was used for nuclear protein extracts. We also conducted pilot experiments using the Stain-Free imaging technology (Bio-Rad) that allows the immediate visualization of proteins at any point during electrophoresis and Western blot to obtain truly quantitative Western blot data by normalizing bands to total protein in each lane. This allowed us to check for the reliability of the loading controls selected in our experimental conditions.

- LOD for the ELISAs need to be presented

Again, this technical information is provided in the datasheet of each ELISA Kit. Data from ELISA were considered reliable if readings were above the limit of detection and in the linearity range of the standard curve.

- Mannitol controls for osmotic effects are mentioned- they should be included in the figures

We preliminarily checked if any effect on HIF-1α activity and cell activation (i.e., the mediator of HG toxicity under investigation and the known outcome of HG (20mM) treatment, respectively) could be observed in endothelial cells and macrophages treated with mannitol 20 mM, as compared to glucose 20 mM (HG). These results are now presented as supplementary file (Supplementary Figure 1) in the revised manuscript.

- The scaling for the y-axis of figures showing the same type of data need to be adjusted to the same level since they are misleading regarding the effect size?

In Graphpad, as in many other graphics software, the axis scale is automatically set based on the data entered. However, as requested, for the same type of data in the same type of cells, we have now set the same scaling for the y-axis (see the revised Figures 1, 3, and 6). Conversely, we did not set the same scaling for the same type of data in different type of cells when pretreatment of U937 cells with LPS had a great impact on the measured target (see Figure 2, panels A – HUVEC – and B – U937 cells). As the Reviewer can appreciate, treatment of U937 cells with LPS alone caused an increase in HIF-1α activity that was greater than that induced by glucose in HUVEC. Setting the y-axis scale in panel A as in panel B would not allow to appreciate the variations induced by the two treatments (i.e., HG and HG+Car), giving a misleading visual message due to lack of discrimination capacity (flattened data at the bottom of the graph). In any case, data from HUVEC and U937 cells are not presented in the same graph. We hope that the Reviewer would agree that her/his request would hamper the understanding of Figure 2 by the journal’s readership without affecting the significance of the results, as assessed using appropriate statistical tests.

- Why are the data presented with SEM instead of SD (beside the obvious effect of making the graphs look nicer?)

There was not additional reason for choosing SEM instead of SD, beside making the graphs easier to read. In fact, as for the y-axis scale, we did not make this choice to convey a misleading message, as the Reviewer seems to suggest. Both SEM and SD are commonly used to describe the variability within the sample in medical journals, and both options are available for data analysis in Graphpad Prism. As the Reviewer knows, the choice between SEM and SD does not impact statistical analysis, the results of which convey the scientific message regardless of the descriptor of variability used.

In general, an overview figure with the conclusion / identified pathway would make it more easy to follow the main story.

We thank the Reviewer for this suggestion. Indeed, with the original submission, we have already provided a graphical abstract that highlights the main findings and pathways involved.

There is no TOC graphics?

As mentioned above, we have already provided a graphical abstract with the original submission; we believe it can be downloaded from the website or requested to the Editor.

Figure 4 A & B & C a rather small and hard to read.

The issue of the figures’ size depends on the final layout of the manuscript. We have provided the individual files of the figures (all with the same size and resolution); again, we believe it can be downloaded from the website or requested to the Editor.

Round 2

Reviewer 2 Report

The authors addressed some, but not all of the requested points. As mentioned in the last review, in general, the manuscript and the study is appreciated. With the comments I intended to improve the scientific standard of the manuscript, which is in my opinion the case. 

I would like to mention here some brief aspects to the authors:

Antibodies, although bought from a company need to be checked for specificity using a blocking peptide, since most of the bought antibodies demonstrate unspecific binding. 

SEM and SD are not just simply to types to represent error bars, they have in a statistical sense different meanings.

Providing cell density is important for the reproducibility since density can have a massive effect on cell growth etc.

Graph pads allows to adjust scaling etc with one simple click using the wand.

In addition, I referred to the MIQUE guidelines regarding the information to be included on PCR, the justification by the authors that this irrelevant information is therefore hard to follow.

I am aware that especially in older published papers, not all required information is provided, but the scientific standards are changing and with that also the information required. As a reviewer I feel obliged to point out those things, at least to point the authors attention to those aspects for future studies.